# Decentralized and Secure Collaborative Framework for Personalized Diabetes Prediction

**DOI:** 10.3390/biomedicines12081916

**Published:** 2024-08-21

**Authors:** Md Rakibul Hasan, Qingrui Li, Utsha Saha, Juan Li

**Affiliations:** Department of Computer Science, North Dakota State University, Fargo, ND 58105, USA; mdrakibul.hasan.3@ndsu.edu (M.R.H.); qingrui.li@ndsu.edu (Q.L.); utsha.saha@ndsu.edu (U.S.)

**Keywords:** diabetes prediction, blockchain, federated learning, machine learning, personalized healthcare

## Abstract

Diabetes is a global epidemic with severe consequences for individuals and healthcare systems. While early and personalized prediction can significantly improve outcomes, traditional centralized prediction models suffer from privacy risks and limited data diversity. This paper introduces a novel framework that integrates blockchain and federated learning to address these challenges. Blockchain provides a secure, decentralized foundation for data management, access control, and auditability. Federated learning enables model training on distributed datasets without compromising patient privacy. This collaborative approach facilitates the development of more robust and personalized diabetes prediction models, leveraging the combined data resources of multiple healthcare institutions. We have performed extensive evaluation experiments and security analyses. The results demonstrate good performance while significantly enhancing privacy and security compared to centralized approaches. Our framework offers a promising solution for the ethical and effective use of healthcare data in diabetes prediction.

## 1. Introduction

Diabetes mellitus is a chronic disease reaching epidemic proportions worldwide. The International Diabetes Federation estimates that over 530 million adults currently live with diabetes, a figure projected to rise dramatically in the coming decades [1]. Uncontrolled diabetes significantly increases the risk of blindness, kidney failure, heart disease, stroke, and lower limb amputation, placing a tremendous burden on individuals, families, and healthcare systems [2].

Early detection and intervention are crucial in mitigating the devastating complications of diabetes [3]. Personalized prediction models have the potential to identify individuals at high risk even before the onset of symptoms. This allows for proactive lifestyle modifications, targeted monitoring, and tailored treatment plans, ultimately leading to improved health outcomes and reduced healthcare costs [4].

Traditional approaches to diabetes prediction often rely on centralized data repositories where vast amounts of patient information are aggregated [5]. While offering the potential for large-scale analysis, these models pose significant challenges. Centralizing sensitive patient data, including medical history, lab results, and lifestyle factors, creates a single point of vulnerability [6]. Data breaches or unauthorized access can have devastating consequences for individuals, leading to potential discrimination, insurance issues, and the misuse of personal health information [7]. The movement of data across networks and its storage in centralized databases further increases these risks [8]. Furthermore, centralized models often rely on datasets sourced from specific hospitals, healthcare systems, or geographical regions. This can introduce biases that limit a model’s generalizability [9]. Variations in demographics, environmental factors, and healthcare practices across different populations may not be adequately represented. Consequently, these models might underperform when applied to individuals or groups outside the original dataset’s scope. 

These limitations raise ethical concerns about patient privacy and highlight the potential for inaccurate or inequitable predictions generated by centralized approaches. The risks of data breaches and the inherent biases within centralized datasets demand a more secure and inclusive approach to diabetes prediction. Moreover, the sensitive nature of healthcare data necessitates strict adherence to privacy regulations such as HIPAA [10] and GDPR [11]. Navigating the complex landscape of data sharing and analysis while ensuring compliance with these regulations presents a significant challenge.

Machine learning offers a powerful toolkit for diabetes prediction [12], with classic algorithms such as support vector machines (SVM) [13] and random forests [14] widely used due to their interpretability and ability to handle mixed data types. More recently, deep learning with neural networks [15] has gained traction, particularly for analyzing large, complex datasets and potentially extracting meaningful patterns from raw health data. Some studies aim to tailor predictions to specific risk factors. For instance, research might focus on the role of genetic markers [16], lifestyle factors, or specific demographic groups for individualized risk assessment.

Research on the use of blockchain in healthcare has rapidly expanded in recent years, demonstrating the technology’s potential to transform various aspects of the healthcare landscape. A key area of focus is the secure storage of electronic health records (EHRs). Blockchain’s decentralization offers resilience against data breaches, while its immutability creates a verifiable audit trail of medical data, crucial for ensuring integrity [17]. Blockchain-powered EHRs can empower patients with greater control over their data, allowing them to grant granular access permissions to different healthcare providers, facilitating seamless interoperability while protecting their privacy [18].

Moreover, blockchain enables patients to selectively share their health information, supporting secure medical research and personalized care [19]. Smart contracts on the blockchain streamline complex healthcare processes by automating data-sharing agreements, payments, and various workflows [20]. Additionally, blockchain’s strengths in tracking provenance are invaluable for ensuring the authenticity and quality control of pharmaceuticals and medical devices, vital for combating counterfeiting and supply chain inefficiencies [21].

Despite these benefits, research also highlights challenges that need to be addressed for widespread blockchain adoption in healthcare. Scalability issues arise due to the need to efficiently store and process large volumes of healthcare data [22]. While blockchain offers enhanced security, healthcare data are subject to stringent privacy regulations like HIPAA. Balancing the need for transparency and auditability on a public blockchain with the requirement to protect patient confidentiality poses a significant challenge [23]. Privacy-preserving techniques and encryption are being explored to address this issue.

To address the limitations of existing diabetes prediction models and bridge the gap between data privacy, regulatory compliance, and collaborative model development, this paper introduces a novel framework that synergistically integrates blockchain and federated learning. While previous studies have explored the individual applications of blockchain or federated learning in healthcare, our approach distinguishes itself by combining these technologies in a cohesive manner to create a decentralized, secure, and privacy-preserving environment for personalized diabetes prediction. 

Furthermore, recognizing the scalability challenges that can arise in federated learning with large-scale networks and massive datasets, our framework employs a decentralized peer-to-peer (P2P) aggregation strategy to mitigate potential bottlenecks and enhance efficiency.

Blockchain as the foundation: Blockchain technology provides a decentralized, immutable, and transparent ledger for secure data management and access control. Patient data can be registered on the blockchain, establishing ownership and enabling fine-grained permissions for how the data can be used. Smart contracts on the blockchain automate data-sharing agreements, ensuring compliance and facilitating auditable transactions between healthcare institutions.Federated learning for privacy preservation: Federated learning allows machine learning models to be trained on distributed datasets without requiring the movement of sensitive patient data. Instead of sending data to a central server, local models are trained on each institution’s dataset. Only model updates, such as gradients or parameters, are exchanged, significantly reducing privacy risks.Blockchain securing federated learning: Blockchain further strengthens federated learning by addressing potential vulnerabilities. It provides a secure channel for model updates, preventing tampering or interception. Additionally, blockchain enables the verification of participating institutions and a transparent audit trail for model training, ensuring trust and accountability within the collaborative network.

This collaborative approach, built on the twin pillars of blockchain and federated learning, enables more robust and personalized prediction models empowered by the combined data resources of multiple healthcare institutions. By preserving privacy and allowing for diverse data sources, the framework aims to overcome the bias and generalization issues inherent in traditional centralized models. Furthermore, our work contributes to the growing body of research exploring the intersection of blockchain, federated learning, and decentralized technologies, with potential implications for areas such as secure data sharing in edge computing and trust management in IoT environments.

## 2. Materials and Methods

### 2.1. System Overview

As shown in Figure 1, our proposed framework establishes a decentralized, privacy-preserving environment for diabetes prediction using federated learning and blockchain technologies. Key components include patients who generate health data, healthcare providers who store and contribute data as nodes on the blockchain network, and the federated learning model itself. A secure blockchain network, utilizing a consensus mechanism like Proof-of-Stake [24], manages data transactions and ensures integrity. Smart contracts on the blockchain enforce data access permissions, automate model exchange, and potentially manage incentives for participation.

Sensitive patient data remains securely stored within each healthcare provider’s local repository. The federated learning model is initialized either on a central server or collaboratively by participants. Local training occurs at each healthcare provider, using only their own data. Encrypted model updates are shared with the blockchain network for secure aggregation, creating an improved global model. The updated model is then distributed for further local refinement. This iterative process emphasizes privacy throughout.

The blockchain provides several security features. It facilitates patient-controlled consent management, preventing unauthorized data use. Smart contracts streamline access verification based on these consents. Additionally, the blockchain authenticates healthcare providers to prevent malicious participation. Its immutable ledger ensures the auditability of all model exchanges and aggregations, aiding in anomaly detection. Secure communication through the blockchain prevents data leakage during the federated learning process.

To address the critical challenge of integrating our framework with diverse Electronic Health Record (EHR) systems, we are leveraging our prior experience in utilizing ontologies and knowledge graphs to achieve interoperability in healthcare data management [25,26,27,28]. By mapping disparate data formats and terminologies onto a standardized schema, we aim to facilitate seamless data exchange and ensure consistent interpretation of healthcare information across different institutions and systems. This approach will be crucial for realizing the full potential of our collaborative framework in real-world healthcare settings.

### 2.2. Privacy Preservation

The foundation of privacy preservation lies in the federated learning model itself. Patient data never leaves the healthcare provider’s local repository. Instead, the machine learning model “travels” to the data for training. Each healthcare provider trains a local copy of the model using their own dataset. This drastically reduces the risk of sensitive data breaches or unauthorized exposure compared to centralized training.

While federated learning provides inherent privacy, additional encryption safeguards are crucial. Sensitive model updates (gradients, weights) exchanged during the aggregation process are encrypted before transmission. Employ strong encryption techniques (e.g., homomorphic encryption [29], secure multi-party computation [30]) to guarantee confidentiality while allowing computations on the encrypted data. Homomorphic encryption allows computations to be performed directly on encrypted data without the need for decryption. In our framework, homomorphic encryption can be used to securely aggregate model updates from different participants without revealing the individual updates. This ensures that sensitive model information remains confidential even during the aggregation process. Secure multi-party computation (MPC) enables multiple parties to jointly compute a function over their inputs while keeping those inputs private. In our framework, MPC is used to perform secure computations on encrypted model updates, ensuring that no single party has access to the raw data or the intermediate results of the computation. 

Beyond these technical safeguards, our framework is designed to align with the core principles of data protection regulations such as HIPAA and GDPR. Patient-controlled consent management, where individuals explicitly grant permission for their data to be used, is a cornerstone of our approach. The blockchain’s immutable ledger ensures that these consents are recorded transparently and cannot be altered without authorization, thus empowering patients and fostering trust. Furthermore, we are actively exploring the integration of data anonymization techniques to further enhance privacy. By removing or modifying personally identifiable information, we can minimize the risk of re-identification while still preserving the utility of the data for model training and analysis.

The blockchain records granular, patient-generated consents as immutable transactions. Smart contracts automatically enforce these consents before granting access to data for model training, research, or other purposes. This puts patients in control and streamlines the consent management process.

Every model exchange, aggregation, and data access request creates a permanent record on the blockchain. This immutable ledger provides a powerful tool for auditing in cases of anomalies or suspected misuse. Additionally, the audit trail can boost transparency and trust in the collaborative system.

### 2.3. Secure Collaboration

Our framework leverages smart contracts on the blockchain to enable secure and efficient collaboration among patients, healthcare providers, and researchers.

#### 2.3.1. Automating Security and Trust

Smart contracts are pivotal in automating and enforcing collaboration agreements among patients, healthcare providers, and researchers. These contracts codify essential terms such as the purpose of data use, duration, compensation, and validation criteria for model updates. By embedding these terms into immutable code, smart contracts ensure adherence to agreed-upon rules, thereby enhancing trust and compliance within the network.

A critical aspect of our smart contracts is the detailed access control mechanism they provide. This system defines who can access patient data, under what conditions, and for what specific purposes. By clearly outlining the roles of data producers (such as healthcare providers) and data consumers (such as researchers or other medical professionals), our framework ensures that only authorized parties can access sensitive health information. This role-based access control is crucial for maintaining patient privacy and data security.

Moreover, smart contracts play a crucial role in automating compliance with regulatory requirements. Data-sharing agreements, which outline the terms and conditions under which patient data can be accessed and used, can be encoded directly into smart contracts. This ensures that any data access or sharing activity is automatically executed in accordance with the predefined rules and regulations, minimizing the risk of non-compliance. Similarly, access control policies can be enforced through smart contracts, granting or revoking access permissions based on predefined criteria and regulatory requirements. This automation not only streamlines the compliance process but also enhances transparency and auditability, as all data access activities are recorded on the blockchain.

The authentication of participants within the network is securely managed through the Ethereum blockchain’s public-key infrastructure. This setup allows for the reliable verification of user identities by matching a participant’s public key with a pre-approved list of keys stored on the blockchain. Following authentication, the smart contracts handle the authorization process, determining the access level of each participant based on their role and the permissions granted to them. This two-step process ensures that sensitive health data are accessed only by verified and authorized entities, thereby preventing unauthorized access and potential data breaches.

To maintain active and continuous participation, our framework employs blockchain-based incentives. By rewarding healthcare providers for their contributions to data sharing and model training, we promote a collaborative spirit. Smart contracts facilitate the transparent and fair distribution of these incentives, ensuring that every contributor is recognized and compensated for their efforts.

We have designed and implemented a suite of smart contracts to manage various aspects of the system. Here we list some important contracts: 

The ModelUpdate smart contract plays a central role in orchestrating the secure and efficient aggregation of model updates from participating healthcare providers in our federated learning framework. Its core responsibilities include:Submission of Model Updates: Healthcare providers, acting as federated learning clients, submit their locally trained model updates (e.g., gradients, weights) to the ModelUpdate contract. These updates are encrypted to ensure data privacy during transmission and storage on the blockchain.Authenticity Verification: The contract verifies the authenticity of each submitted update using digital signatures. Each participating healthcare provider has a unique cryptographic key pair, and the update is signed using their private key. The contract then uses the provider’s public key to verify the signature, ensuring that the update originated from an authorized participant.Validation and Integrity Checks: Before accepting an update, the contract performs various validation and integrity checks to ensure the quality and trustworthiness of the submitted model. These checks may include:
(1)Data Validation: Verifying that the update adheres to the expected data format and structure.(2)Model Performance Metrics: Checking if the update meets certain performance criteria or thresholds to prevent the inclusion of potentially harmful or inaccurate models.(3)Dataset Verification: Ensuring that the update was trained on an authorized and relevant dataset, preventing the inclusion of models trained on biased or irrelevant data.Secure Aggregation: Once the updates are validated, the contract employs secure aggregation mechanisms to combine them into a new global model. This aggregation process may utilize techniques such as:(1)Federated Averaging: A simple and widely used approach that averages the model updates from different participants, weighted by the size of their local datasets.(2)Secure Multi-Party Computation (MPC): Enables the computation of the aggregated model without revealing the individual updates, ensuring data privacy.(3)Homomorphic Encryption: Allows computations to be performed directly on encrypted data, further enhancing privacy during aggregation.Consensus Mechanisms: The contract leverages the underlying blockchain’s consensus mechanism (e.g., Proof-of-Stake) to ensure agreement among network participants on the final aggregated model. This prevents malicious actors from manipulating the model or introducing biased updates.Model Distribution: After successful aggregation, the updated global model is securely distributed back to the participating healthcare providers, enabling them to continue local training and further refine the model in subsequent iterations.

Data Access Control: The DataAccessControl contract manages patient consent and data access permissions. It enforces fine-grained access control based on patient preferences and regulatory requirements, ensuring that data are used only for authorized purposes.

Incentive contract: The Incentive contract handles the distribution of rewards to healthcare providers for their contributions to data sharing and model training. It uses transparent and auditable logic to calculate and distribute incentives fairly, promoting active participation in the network.

These smart contracts are implemented using the Solidity programming language and deployed on the Ethereum blockchain. We have carefully designed the contract logic to be gas-efficient and secure, minimizing transaction costs and potential vulnerabilities. Detailed examples of smart contracts can be found in Section 3.3.2. Blockchain Smart Contract Evaluation with Use Cases.

#### 2.3.2. Enhancing Security in Federated Learning

In addition to fostering collaboration, our framework emphasizes the security of the federated learning process, employing blockchain technology to protect data integrity and participant privacy.

Model Authentication: Blockchain verifies the identities of healthcare providers participating in federated learning, mitigating the risk of malicious actors or corrupted model updates.Secure Parameter Aggregation: Smart contracts provide a safe channel for model update exchange and aggregation. Advanced techniques like secure multi-party computation or homomorphic encryption can be integrated into the smart contract logic to ensure the confidentiality of updates during aggregation.Auditability and Quality Control: Storing model updates, metadata (e.g., hyperparameters, dataset origins), and aggregation outcomes on the blockchain creates an immutable audit trail. This promotes transparency and facilitates the investigation of anomalies. Smart contracts can enforce validation checks on model updates (for accuracy, fairness, or adherence to agreed-upon datasets) before they are accepted into the global model.

#### 2.3.3. Empowering Patients with Data Sovereignty

The framework seamlessly integrates patient-generated health data from wearables and personal devices. Blockchain technology empowers patients to directly contribute to research with granular control over data permissions. Smart contracts ensure that patient data are used solely in accordance with their preferences, driving personalized model development.

### 2.4. Personalized Diabetes Prediction 

In the development of our personalized diabetes prediction model, the choice of machine learning techniques plays a pivotal role and is largely dictated by the characteristics of the data at hand. For instance, time series data such as continuous glucose monitoring (CGM) readings or sequential health records necessitate models like Recurrent Neural Networks (RNNs) [31] and Long Short-Term Memory Networks (LSTMs) [32], which are adept at capturing temporal patterns and long-range dependencies, respectively. These models are particularly useful for analyzing trends in glucose levels over time. For structured data, which includes demographics, lab results, and medication logs presented in tabular form, models like Decision Trees [33] or Random Forests [34] are preferred for their interpretability and ability to handle diverse data types. Support Vector Machines (SVMs) [35] are another option, known for their efficacy in classification tasks and capability to manage high-dimensional spaces, making them suitable for complex datasets. When it comes to medical image analysis, such as identifying signs of diabetic retinopathy in retinal scans, Convolutional Neural Networks (CNNs) [36] are the go-to choice. These deep learning models are specifically designed for image processing, enabling them to extract intricate features that are indicative of diabetes-related complications.

The federated learning process underpins the personalization aspect of the model. Initially, a global model is either trained on a small, curated dataset that complies with privacy regulations or is collaboratively initialized by participants sharing secure parameters. Subsequently, each healthcare provider enhances this global model with their local data, contributing only encrypted model updates back to the network, thereby safeguarding patient privacy.

The aggregation of these updates into a refined global model is managed through smart contracts, ensuring a secure and verifiable process. These contracts not only receive the encrypted updates but also oversee the execution of secure aggregation methods, like federated averaging, to integrate the updates into an improved model. This model is then redistributed to all participants for further local tuning, with the cycle of updates and aggregation continuing until the model converges or reaches a predetermined number of iterations.

The strength of federated learning lies in its ability to learn from the vast and varied data across multiple healthcare providers, enhancing the robustness of the global model. This diversity enables the model to make more accurate predictions across different patient demographics, treatment regimens, and risk factors. Moreover, the analysis of aggregated model updates can uncover unique patient subgroups, offering insights into tailored treatment approaches and contributing to more personalized diabetes care in the future.

### 2.5. Scalability Considerations 

Blockchain networks, particularly public ones like Ethereum, face inherent scalability challenges due to their decentralized nature and consensus mechanisms. These challenges become especially critical in the context of healthcare applications, where the handling of large datasets and the need for frequent transactions are commonplace. Ethereum’s limitations, such as high gas fees and limited transaction throughput, can significantly impact the practicality and cost-effectiveness of any blockchain-based solution.

In our framework, we have taken a proactive approach to address these scalability concerns through careful design choices and targeted technical implementations. One key strategy involves leveraging the InterPlanetary File System (IPFS) for off-chain data storage. IPFS, a decentralized and peer-to-peer file storage system, allows for efficient and secure management of large datasets. In our implementation, we utilize IPFS to store sensitive patient data and computationally intensive model updates, thereby reducing the burden on the Ethereum blockchain. Only the corresponding hashes and essential metadata are recorded on-chain, significantly decreasing the on-chain data footprint and transaction volume, ultimately leading to improved scalability and reduced gas fees.

Furthermore, we have meticulously designed our smart contracts with a focus on gas efficiency and minimizing unnecessary computations. This optimization involves utilizing efficient data structures like mappings and arrays to streamline the storage and retrieval of information within the contract, thus reducing storage costs and enhancing execution speed. We have also employed optimized algorithms for critical functions such as data validation, access control, and model aggregation, aiming to minimize gas consumption during contract execution. Additionally, we have adopted a modular approach by breaking down complex operations into smaller, reusable functions. This enhances code readability and reduces the overall contract size, thereby lowering both deployment and execution costs.

Crucially, our federated learning process employs a decentralized, peer-to-peer (P2P) model aggregation strategy. This approach eliminates the reliance on a central server, which can become a bottleneck in traditional federated learning implementations. By enabling direct communication and aggregation of model updates among participating nodes, we distribute the computational and communication load, enhancing the overall scalability and fault tolerance of the system. This P2P approach also allows for parallel processing of model updates, further improving efficiency as the number of participants and data volume increase.

As will be further elaborated in the future work section, we are also committed to exploring additional scalability-enhancing mechanisms, such as batching, aggregation, and the adoption of alternative consensus mechanisms, to ensure our framework’s continued adaptability and effectiveness in the face of evolving blockchain technologies and healthcare data demands.

## 3. Results

Our study leverages a publicly available diabetes prediction dataset, which we deploy on the Ethereum blockchain [37], to simulate realistic scenarios of data sharing and prediction in a decentralized healthcare context. This approach allows us to not only explore the practical applications of blockchain technology in managing sensitive health data but also rigorously evaluate the performance, privacy, and security aspects of our proposed framework.

### 3.1. Experimental Setup

#### 3.1.1. Blockchain Network

Platform: We utilized the Ethereum blockchain as the foundation for our decentralized network due to its mature ecosystem, extensive developer tools, and support for smart contracts.Nodes: We simulated a network of five participating nodes, each representing a healthcare provider or institution, to emulate a collaborative environment for data sharing and model training.Smart Contracts: We developed and deployed a suite of smart contracts on the Ethereum blockchain using the Solidity programming language. These contracts handle key functionalities such as user registration, authentication, data access control, model update management, and incentive mechanisms.Development Environment: The smart contracts were developed and tested using the Remix Integrated Development Environment (IDE) [38], which provides a user-friendly interface for interacting with and deploying smart contracts on the Ethereum network.

#### 3.1.2. Dataset

Source: We employed the publicly available diabetes prediction dataset [39], which comprises medical and demographic data from patients along with their diabetes status (positive or negative).Data Partitioning: To simulate a distributed data scenario, we partitioned the dataset into five distinct chunks, each assigned to a different node on the blockchain network. This emulates the real-world situation where healthcare providers hold their own portions of patient data.Data Preprocessing: Prior to model training, we performed comprehensive data preprocessing steps to ensure data quality and consistency:(1)Missing Value Handling: Missing values were addressed using imputation techniques [40] (e.g., mean/median replacement) or deletion based on the extent and distribution of missingness.(2)Categorical Feature Encoding: Categorical features like gender and smoking history were encoded using one-hot encoding [41] or ordinal encoding [42].(3)Normalization and Scaling: Continuous features such as BMI and HbA1c levels were normalized and scaled to enhance model stability and convergence.(4)Class Imbalance Handling: To address the class imbalance in the dataset (fewer positive diabetes cases), we applied undersampling techniques [43] to create a more balanced distribution.

#### 3.1.3. Model Selection and Training

Algorithm: We chose the XGBoost algorithm [44] as our primary model due to its proven effectiveness in handling structured healthcare data and its ability to manage complex feature interactions.Hyperparameter Optimization: We utilized Optuna [45], a hyperparameter optimization framework, to fine-tune the XGBoost model and identify the most effective combination of hyperparameters to maximize performance metrics, such as the F1 score.

### 3.2. Blockchain-Based Federated Learning Workflow

To replicate a real-world scenario of distributed healthcare data, we strategically partition the diabetes dataset into chunks. Each chunk is then deployed to a distinct node within our blockchain network. This simulates a scenario where diverse healthcare providers securely store their own portions of patient data.

Before initiating federated learning, we train a base global model using a small portion of the dataset. This initial model serves as a common starting point for all participants in the federated learning process. Each node on the blockchain network receives a copy of the initialized global model. This marks the beginning of the federated learning process.

Each node, acting as a healthcare provider, fine-tunes the global model using the locally stored chunk of the dataset. Importantly, only the model updates (gradients or parameters) are shared with the blockchain, ensuring the raw patient data remains protected within each node. The blockchain network uses the smart contract to securely aggregate the model updates received from all nodes. The updated global model is distributed back to all nodes in the network through a smart contract. These steps are repeated until the global model achieves the desired performance level.

### 3.3. Evaluation Results

#### 3.3.1. Federated Learning Model Performance

Metrics: We evaluated the performance of the federated learning model on an independent test set using key metrics, including accuracy, precision, recall, and F1-score.Convergence: We observed that the accuracy, precision, recall, and F1-score of the individual client models improved over the training rounds, converging towards stable and high performance after approximately four rounds (Figure 2, Figure 3, Figure 4 and Figure 5).Comparison with Centralized Learning: We also compared the communication overhead of our federated learning approach with that of centralized learning, demonstrating a significant reduction in communication costs (Figure 6).

We initiated a global XGBoost model, distributing it to five participating clients for training on their respective local datasets. The cornerstone of our federated learning approach lies in aggregating these individually trained models back into a unified global model. This aggregation is achieved by retraining the global model on a compilation of predictions from each local model, effectively synthesizing diverse learnings into a cohesive whole.

To refine and enhance the global model, we employed an iterative process of sending the model back to each client for further training, followed by the aggregation step. This cycle was repeated over 10 rounds, aiming to progressively improve the model’s predictive performance. Upon completion, the federated learning model’s efficacy was rigorously evaluated on an independent test set, focusing on accuracy, precision, recall, and F1-score to assess its generalization capabilities.

As depicted in Figure 2, the accuracy of each client’s model improved over the training rounds, converging towards a stable and high performance after approximately four rounds. This demonstrates the effectiveness of the federated learning approach in collaboratively enhancing the model’s ability to correctly classify instances of diabetes.

**Figure 2 biomedicines-12-01916-f002:**
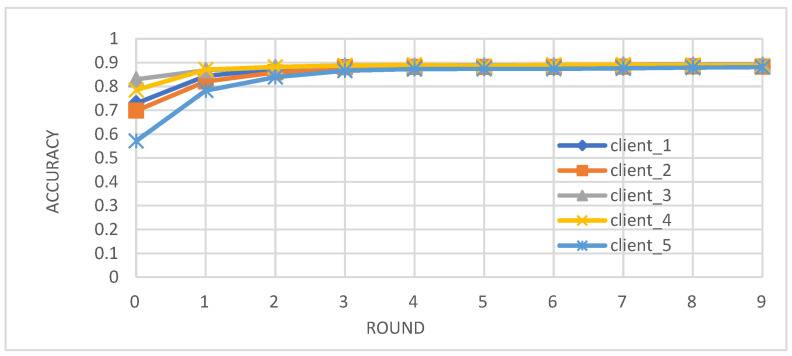
Accuracy of individual client learners over time.

Figure 3 illustrates the precision evolution across the training rounds. The precision of each client improves as training progresses. This indicates the model’s increasing ability to minimize false positive predictions.

**Figure 3 biomedicines-12-01916-f003:**
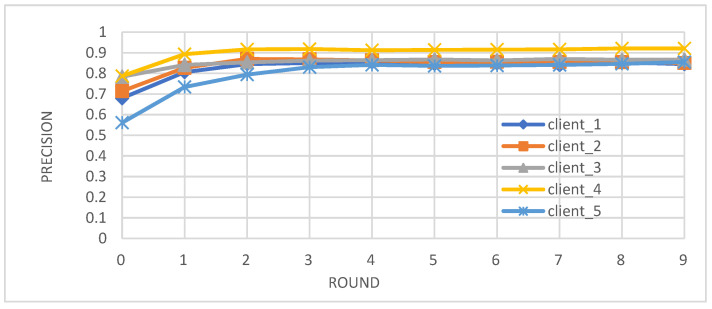
Precision of individual client learners over time.

In Figure 4, we observe a consistent upward trend in recall for each client, highlighting the model’s enhanced ability to identify true positive cases and reduce the occurrence of false negatives. The recall values largely converge after the fourth round.

**Figure 4 biomedicines-12-01916-f004:**
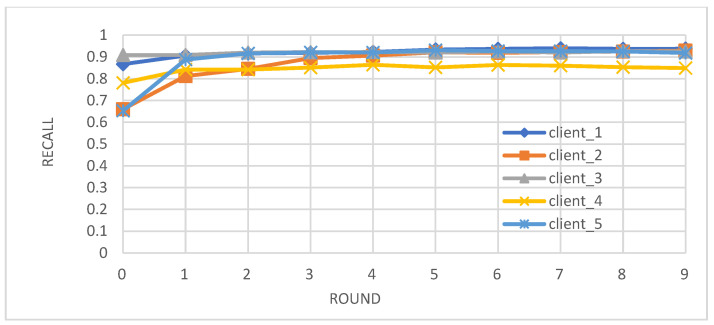
Recall of individual client learners over time.

The F1-score, a balanced metric combining precision and recall, showcases a steady improvement across training rounds in Figure 5. Notably, the F1 scores for all clients reach a stable point after the fourth round, reflecting the model’s growing capability to strike an optimal balance between minimizing false positives and false negatives, ultimately leading to a more robust and reliable diabetes prediction model.

**Figure 5 biomedicines-12-01916-f005:**
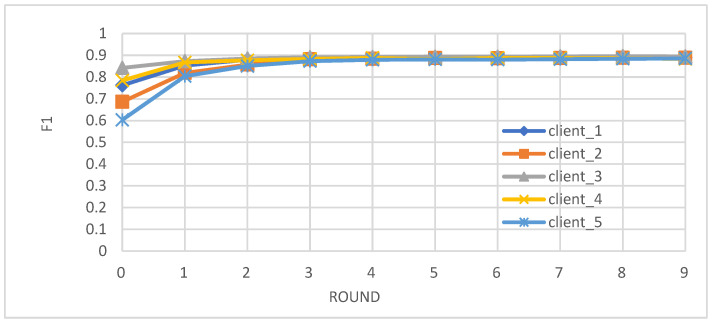
F1-score of individual client learners over time.

In conclusion, our decentralized, federated learning approach successfully demonstrates the potential of collaborative learning in enhancing the performance of diabetes prediction models. The iterative training and aggregation process led to significant improvements in accuracy, precision, recall, and F1-score, with convergence observed after approximately four rounds. This highlights the power of harnessing diverse data sources while preserving privacy.

A significant advantage of federated learning over centralized approaches lies in its dramatically reduced communication overhead. In centralized learning, the entirety of the raw data from all clients must be transmitted to a central server for model training. This process not only incurs substantial communication costs, especially when dealing with large and complex datasets, but also raises significant privacy concerns as sensitive patient information is exposed during transmission. In contrast, our federated learning framework drastically minimizes communication overhead by only transmitting model parameters or updates between the clients and the central server. These updates are typically much smaller in size compared to the raw data, resulting in significantly reduced communication requirements. This efficiency is particularly crucial in healthcare settings, where bandwidth limitations and privacy regulations may pose challenges to data transfer.

Figure 6 visually illustrates the stark difference in communication overhead between centralized and federated learning. The bar represents centralized learning towers over those of federated learning, emphasizing the substantial communication savings achieved by the latter. This reduction in communication not only improves the efficiency and scalability of the training process but also enhances privacy by minimizing the exposure of sensitive data during transmission. Furthermore, the decentralized nature of federated learning eliminates the need for a single point of data collection, further mitigating the risk of data breaches and unauthorized access. By keeping the data localized at each client, our approach ensures that patient privacy is maintained while still enabling collaborative model training.

**Figure 6 biomedicines-12-01916-f006:**
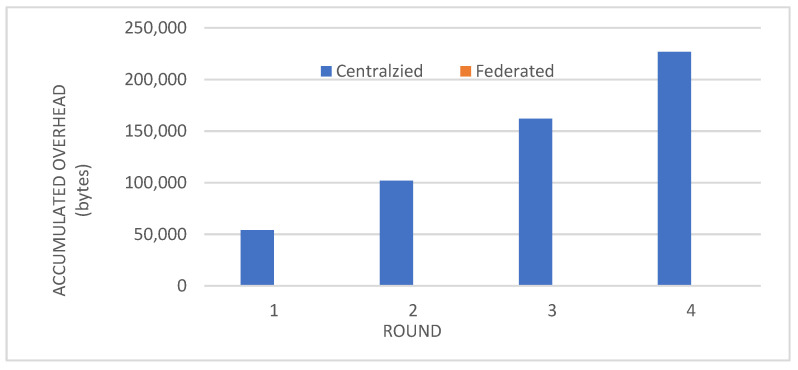
Communication overhead between centralized and federated learning.

In summary, the reduced communication overhead of federated learning, as clearly demonstrated in Figure 6, makes it a highly attractive solution for privacy-conscious and resource-constrained healthcare environments. By minimizing the amount of data transmitted and avoiding the need for centralized data storage, our framework offers a more efficient, secure, and scalable approach to diabetes prediction.

#### 3.3.2. Blockchain Smart Contract Evaluation with Use Cases

We showcased the effectiveness of our smart contracts in enhancing data privacy and security through a series of use cases, illustrating their functionalities for user registration, authentication, data access control, secure model aggregation, and malicious update prevention (Figure 7, Figure 8, Figure 9, Figure 10, Figure 11 and Figure 12). 

Our implementation of the Ethereum blockchain allowed us to rigorously assess how smart contracts enhance data privacy and security within the healthcare system. We present the following use cases, each illustrated with screenshots of smart contract execution results, to demonstrate the effectiveness of our approach.

Data Owner Registration

Nora, a patient, has chosen to utilize our blockchain-based application for the secure and decentralized management of her health records. By registering on the platform with her Ethereum address (0x5B3....eddC4), Nora establishes a unique identifier that ensures all her health data generated during hospital visits is securely stored and exclusively accessible through the application.

**Figure 7 biomedicines-12-01916-f007:**
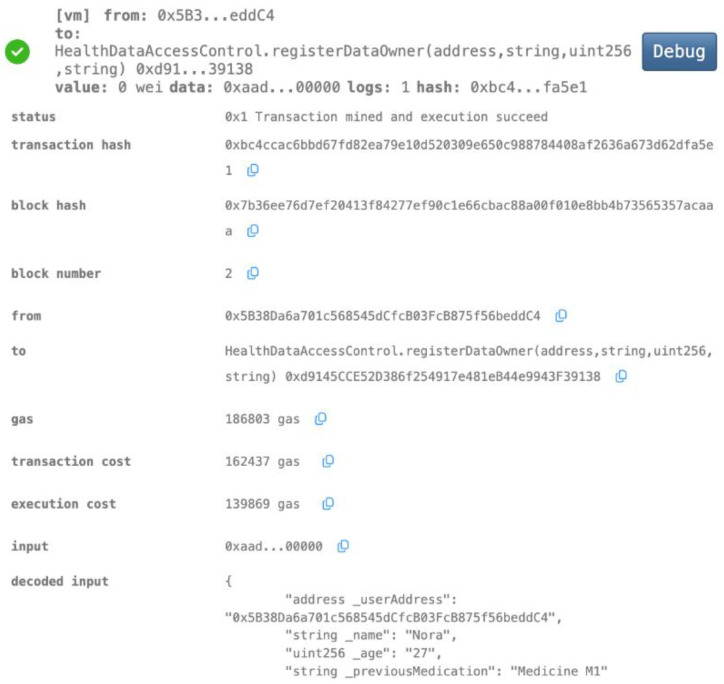
A new patient (Data Owner) is registered.

Granting Access to Healthcare Providers

Nora wishes to grant her new healthcare provider, Dr. Alex (Ethereum address 0xAb8....35cb2), access to her health records. She registers Dr. Alex in the system and assigns him the role of a healthcare practitioner (producer). Figure 8a,b demonstrate the corresponding smart contract and the execution log. The smart contract successfully registers Dr. Alex with the designated role and establishes a link between his Ethereum address and Nora’s health data. The transaction receipt confirms the successful execution, and the contract’s internal state reflects the granted access permissions. This ensures that Dr. Alex can securely access and potentially contribute to Nora’s health data within the platform. 

**Figure 8 biomedicines-12-01916-f008:**
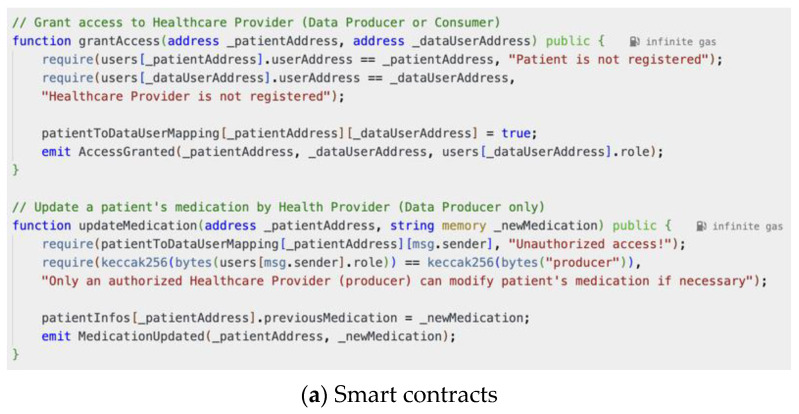
Data Owner (Nora) has granted her patient data access to Data Producer (Dr. Alex).

Revoking Access

Nora decides to revoke Dr. Alex’s access to her health records due to a change in healthcare providers or privacy concerns. Utilizing the smart contract’s RevokeAccess() function, Nora successfully removes Dr. Alex’s permissions to her health data. As shown in Figure 9, the transaction receipt confirms the execution, and the contract’s internal state is updated to reflect the revoked access. This ensures that Dr. Alex can no longer view or modify Nora’s health information, upholding her control and autonomy over her personal data.

**Figure 9 biomedicines-12-01916-f009:**
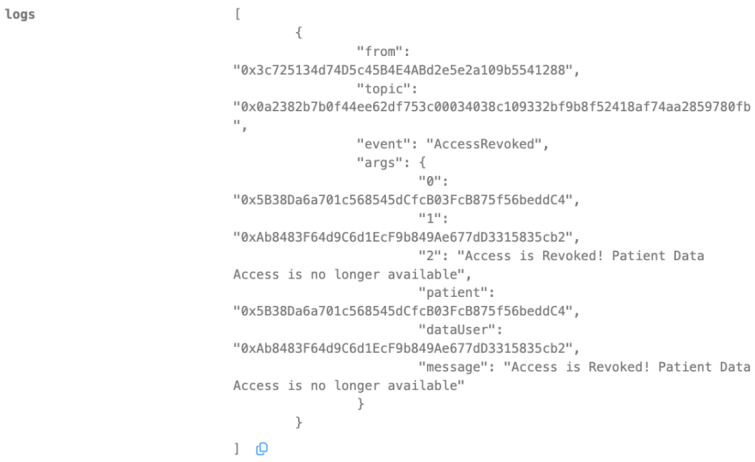
Access has been revoked by the data owner.

Unauthorized Access Prevention

A malicious actor, not registered in the system (e.g., 0xdD8….92148), attempts to gain unauthorized access to Nora’s health data or modify her medication records, such as previous medication. The smart contract’s access control mechanism detects that the requesting entity lacks the necessary permissions. The transaction is automatically denied, returning a “false” state. Any attempted changes are reverted, ensuring the integrity and confidentiality of Nora’s health information. This demonstrates the system’s robust protection against unauthorized intrusion. Only if the actor gets authorized (then he must be a producer) can he modify the patient’s (Nora) medication records.

**Figure 10 biomedicines-12-01916-f010:**
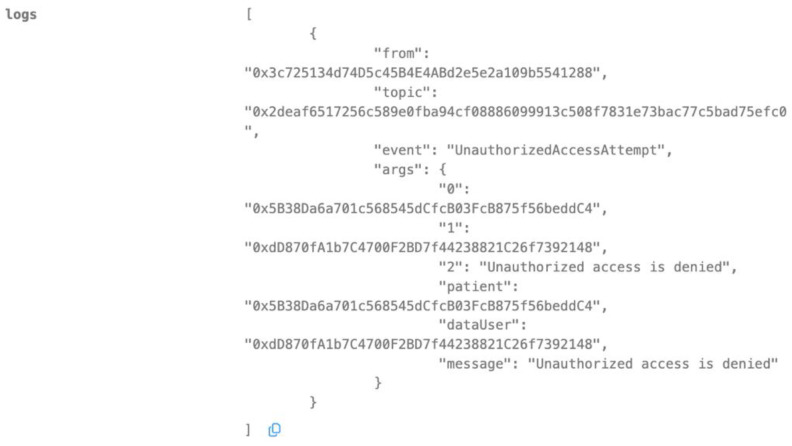
Access from an unauthorized entity is denied.

Secure Model Aggregation of Federated Learning

Multiple authorized FL clients, such as hospitals and research centers, submit their locally trained models to the blockchain. The smart contract verifies the authenticity and integrity of each submitted model, ensuring that only updates from authorized participants are included in the aggregation process. As shown in Figure 11, after several rounds of secure aggregation, the global model’s accuracy reaches 86%. This demonstrates the smart contract’s crucial role in maintaining the security and validity of the federated learning process, ultimately leading to a more accurate and reliable model without compromising data privacy.

**Figure 11 biomedicines-12-01916-f011:**
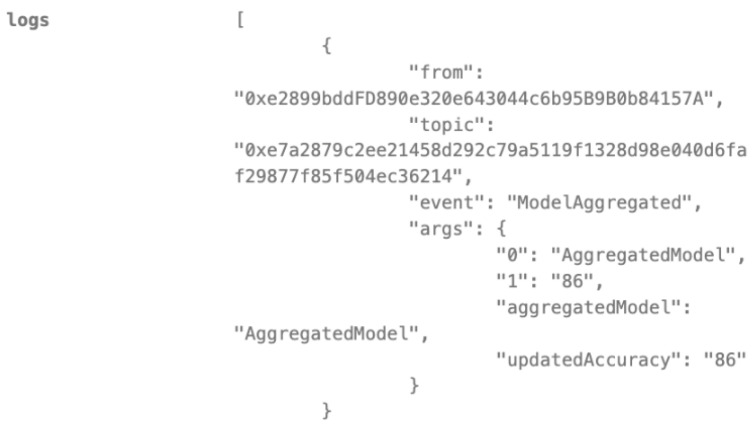
Local models are aggregated into global models.

Mitigating Malicious Model Updates of Federated Learning

In a simulated attack, a compromised FL client attempts to inject malicious model updates into the federated learning process. The smart contract’s security mechanisms, including data validation and integrity checks, successfully detect malicious updates. As shown in Figure 12, a non-registered FL Client (0xdD8….92148) tried to submit a malicious update. The smart contract rejects the particular model update and reverts the transaction to the initial state since only registered participants can perform the model submission. The contract rejects the submission and prevents the compromised model from influencing the global model. This demonstrates the effectiveness of blockchain-based defenses in maintaining the integrity and trustworthiness of the federated learning process, even in the face of potential threats.

**Figure 12 biomedicines-12-01916-f012:**
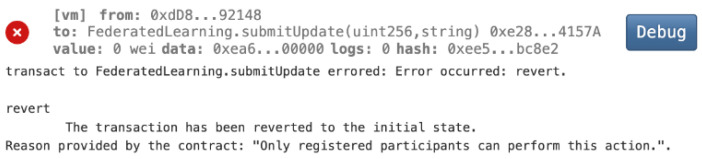
Malicious model updates are avoided.

## 4. Discussion

Our study demonstrates the successful integration of blockchain technology and federated learning (FL) for privacy-preserving diabetes prediction. By leveraging the decentralized nature of blockchain and the collaborative learning capabilities of FL, we achieved significant improvements in model performance while safeguarding sensitive patient data. The key findings and interpretations are summarized as follows:(1)Enhanced Model Performance: The federated learning approach, as evidenced by the convergence of accuracy, precision, recall, and F1-score after just a few rounds (Figure 2, Figure 3, Figure 4 and Figure 5), demonstrates the power of collaborative learning from diverse data sources. This suggests that the model can effectively generalize to new, unseen data, making it a promising tool for real-world diabetes prediction.(2)Robust Privacy and Security: The implementation of smart contracts on the Ethereum blockchain ensured the secure management of patient data throughout the process. The use cases illustrated the efficacy of access control mechanisms in preventing unauthorized access and modification attempts (Figure 7, Figure 8, Figure 9 and Figure 10). Additionally, the secure aggregation of local models in federated learning further reinforced data privacy.(3)Reduced Communication Overhead: The significant reduction in communication overhead compared to centralized learning (Figure 6) not only improves efficiency but also minimizes potential data breaches. This is particularly important in healthcare settings, where privacy regulations and bandwidth limitations are critical considerations.

Our blockchain-powered collaborative framework significantly enhances security and privacy in diabetes prediction. By keeping patient data decentralized and utilizing federated learning, we inherently minimize the risk of data breaches and unauthorized access. Additionally, the integration of smart contracts on the Ethereum blockchain provides robust defenses against various threats:(1)Secure model sharing: Blockchain’s immutable ledger ensures a transparent and tamper-proof record of model updates, safeguarding against unauthorized modifications.(2)Malicious update prevention: Simulated attacks, such as the injection of malicious model updates, were successfully thwarted by our smart contract’s data validation and integrity checks.(3)Access control: Smart contracts enforce strict authorization and access control, using token-based mechanisms and Solidity modifiers to prevent unauthorized actions and mitigate reentrancy attacks.(4)Transparency and traceability: The blockchain ledger’s transparency enables the tracking of all model updates and data access requests, ensuring auditability and accountability.(5)DoS mitigation: The Ethereum network’s gas fee requirement naturally limits the number of requests a single source can make, effectively mitigating Denial of Service (DoS) attacks.(6)MITM and replay attack prevention: Token-based authentication with private/public key verification secures communications within the network, preventing Man-in-the-Middle (MITM) and replay attacks.(7)Input validation and sanitization: Smart contracts rigorously validate and sanitize all incoming data, including model updates and patient consent, to prevent malicious data injection or manipulation.(8)Anomaly detection: The blockchain’s immutable ledger and transparent transaction history enable the implementation of anomaly detection mechanisms to identify and address suspicious activities or potential security breaches.(9)Blockchain-specific attack mitigation: We have considered potential blockchain-specific attacks such as 51% attacks and double-spending. Our framework leverages the security features of the underlying blockchain network and employs additional safeguards within the smart contract logic to mitigate these risks.

These multifaceted security measures, combined with the inherent privacy advantages of federated learning, establish our framework as a trustworthy and reliable solution for diabetes prediction in the healthcare domain.

## 5. Conclusions

This study presents a novel framework that combines blockchain technology and federated learning to address the critical challenges of privacy, security, and data diversity in diabetes prediction. Our approach leverages the strengths of both technologies to create a decentralized, secure, and collaborative environment for developing personalized prediction models.

The evaluation results demonstrate the effectiveness of our framework in achieving high predictive performance while preserving patient privacy. The federated learning approach, with its iterative model training and aggregation, significantly improved model accuracy, precision, recall, and F1 score. The smart contract-based system on the Ethereum blockchain ensured secure data management, access control, and model exchange, safeguarding sensitive patient information throughout the process.

Furthermore, our framework’s robust security measures, including data validation, access controls, and blockchain-based authentication, effectively mitigated various potential threats, such as unauthorized access, malicious model updates, and denial-of-service attacks. The immutability and transparency of the blockchain ledger provided an additional layer of trust and accountability, further enhancing the overall security of the system.

The successful implementation of our framework using a real-world diabetes dataset showcases the practicality and potential impact of this approach in the healthcare domain. By enabling secure and collaborative model training without compromising data privacy, we pave the way for the development of more accurate and personalized diabetes prediction models, ultimately leading to improved patient outcomes and healthcare delivery.

However, challenges such as scalability, integration with existing healthcare systems, and evolving regulatory frameworks need to be addressed for the widespread adoption of blockchain-based federated learning in healthcare. Future research should focus on optimizing the scalability of blockchain networks, developing standardized data models and ontologies, and establishing clear legal frameworks to ensure compliance with privacy regulations and data protection laws.

In particular, we recognize the need to further enhance the scalability of our framework to accommodate the growing volume of healthcare data and the increasing number of participants in federated learning networks. Future work will investigate the implementation of advanced scalability-enhancing mechanisms, such as:Batching and Aggregation: Combining multiple model updates or data transactions into a single transaction to reduce the overall number of on-chain transactions and associated gas fees.Alternative Consensus Mechanisms: Exploring consensus mechanisms like Proof-of-Stake (PoS) or Delegated Proof-of-Stake (DPoS) that offer improved scalability compared to Ethereum’s current Proof-of-Work (PoW) mechanism.

By addressing these challenges and incorporating these advancements, we aim to create a truly scalable, secure, and privacy-preserving framework that can revolutionize the way healthcare data are shared, analyzed, and utilized for personalized and preventive care.

In conclusion, our proposed framework represents a significant step towards a more secure, privacy-conscious, and collaborative future for healthcare data sharing and analysis. By harnessing the power of blockchain and federated learning, we have demonstrated a promising pathway for unlocking the full potential of healthcare data while upholding ethical and regulatory standards, paving the way for a new era of personalized and preventive healthcare.

## Figures and Tables

**Figure 1 biomedicines-12-01916-f001:**
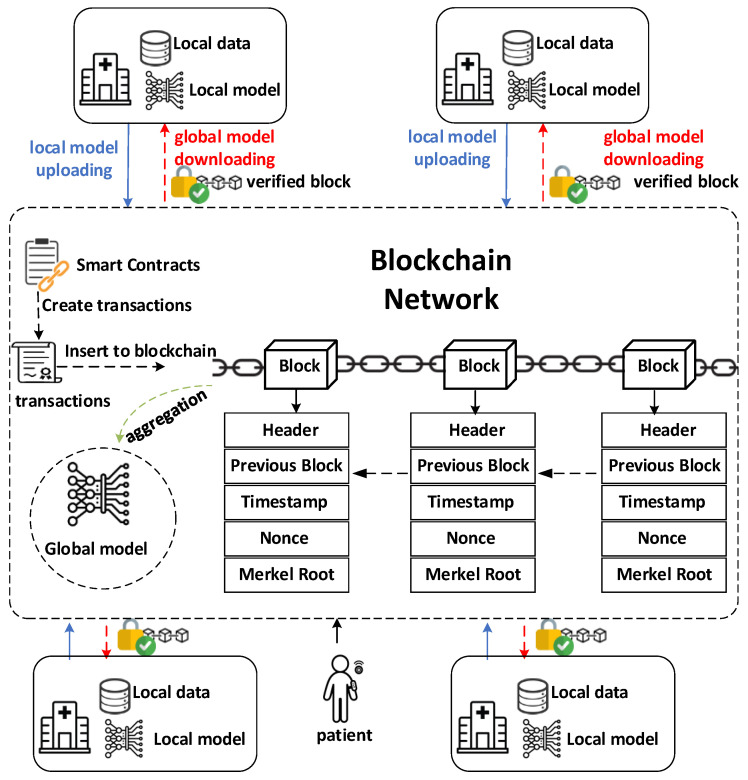
System framework.

## Data Availability

The diabetes prediction dataset can be found at: https://www.kaggle.com/datasets/iammustafatz/diabetes-prediction-dataset/data (accessed on 20 August 2024).

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
