# Peer review of "Decentralized and Secure Collaborative Framework for Personalized Diabetes Prediction"

_biomedicines, 2024, doi:10.3390/biomedicines12081916_

Round 1
Reviewer 1 Report
Comments and Suggestions for Authors
The authors have well demonstrated the effectiveness and robustness of their proposed framework. However, I'd suggest they improve their manuscript on the following areas:
1. please address the challenges related to the scalability of blockchain networks (how would they manage Ethereum's limitations, such as gas fee and transaction throughput)
2. Please consider discussing evolving regulatory frameworks and how to establish legal guidelines for ensuring compliance with privacy regulations and data protection laws.
3. They need to discuss in depth how the federated learning approach scales with an increasing number of participants and data size.
4. There is no discussion on Integration challenges, such as (a) Interoperability (interaction of the proposed framework with different EHR systems); (b) data standards (just mentioning about the integration issue with existing healthcare systems is not enough, the authors need to discuss the need for standard data models and ontologies to ensure seamless integration)
5. Implementation mechanisms are mentioned but not discussed properly. Therefore, a detailed description of the security measures (such as encryption techniques, threat mitigation strategies, access control, etc.) and smart contracts (how the design and implementation of smart contracts for model updates, data access control, incentive mechanisms, etc. will be carried out) needs to be provided.
6. Some sections miss clear explanations and need a more structured presentation of the methodologies and findings, such as (a) the introduction section: fails to establish the novelity of their proposed framework compared to the existing state of the art; (b) the methodology section: needs to provide a step-by-step explanation of the implementation of the framework. The most important missing components are: specific algorithms, data preprocessing steps, model training processes, and the exact role of federated learning and blockchain in the proposed framework, to name a few.
Author Response
- please address the challenges related to the scalability of blockchain networks (how would they manage Ethereum's limitations, such as gas fee and transaction throughput)
Response: We acknowledge that the initial manuscript did not explicitly address the scalability challenges in sufficient detail. We have taken this feedback into serious consideration and have revised our paper to provide a more comprehensive discussion of the scalability concerns and our proposed solutions.
To address this concern, we have made the following revisions to our paper:
In Section 2 Materials and Methods: We have added a new subsection 2.5 dedicated to "Scalability Considerations." In this subsection, we explicitly discuss the potential scalability limitations of blockchain networks, particularly within the context of Ethereum and its gas fee and transaction throughput constraints. We detail the specific strategies we have implemented in our current prototype to mitigate these challenges, including off-chain data storage using IPFS and optimized smart contract design.
In Section 5 Conclusion: We have highlighted our future research directions, emphasizing the exploration of additional scalability-enhancing mechanisms such as batching, aggregation, and alternative consensus mechanisms.
We believe these revisions provide a more transparent and comprehensive overview of our approach to addressing the scalability challenges inherent in blockchain-based healthcare applications. We are committed to continuously refining and improving our framework to ensure its practicality and effectiveness in real-world healthcare settings.
- Please consider discussing evolving regulatory frameworks and how to establish legal guidelines for ensuring compliance with privacy regulations and data protection laws.
Response: We appreciate the reviewer's emphasis on the importance of addressing evolving regulatory frameworks and compliance with privacy regulations. We recognize that the dynamic nature of these legal landscapes presents a crucial challenge for the widespread adoption of blockchain-based federated learning in healthcare.
To address this concern, we propose the following revisions throughout the manuscript:
In Section 1 Introduction, we acknowledge the regulatory challenges associated with healthcare data sharing and analysis, highlighting the need for compliance with privacy regulations such as HIPAA and GDPR.
In Section 2 Materials and Methods’ subsection 2.2 Privacy Preservation, we expand the discussion on privacy preservation to include the role of regulatory compliance. Highlight how the framework's design principles, such as patient-controlled consent management and data anonymization techniques, align with the requirements of data protection laws. In subsection 2.3 Secure Collaboration, we discuss the potential role of smart contracts in automating compliance with regulatory requirements, such as data sharing agreements and access control policies.
- They need to discuss in depth how the federated learning approach scales with an increasing number of participants and data size.
Response: We appreciate the reviewer's insightful comment regarding the scalability of our federated learning approach with an increasing number of participants and data size. We want to emphasize that our blockchain-based framework already employs a decentralized, peer-to-peer (P2P) aggregation strategy, which inherently addresses the single point of failure and bottleneck issues often associated with centralized federated learning approaches.
In our implementation, model updates are securely exchanged and aggregated directly among participating nodes on the blockchain network, eliminating the reliance on a central server. This decentralized approach not only enhances the robustness and fault tolerance of the system but also contributes to improved scalability by distributing the computational and communication load across multiple nodes. Moreover, the scalability-enhancing strategies we have discussed, such as off-chain data storage and optimized smart contract design further contribute to addressing the challenges of handling a growing number of participants and larger datasets. These techniques aim to optimize communication, computation, and storage resources, ensuring that our federated learning approach remains efficient and effective even as the network expands and the volume of healthcare data increases.
We acknowledge that further research and development are needed to explore additional scalability-enhancing mechanisms, such as hierarchical federated learning, model compression, and adaptive aggregation, to ensure the framework's continued adaptability to larger and more complex healthcare applications.
To further clarify this issue, we have revised the manuscripts in the following ways:
In Section 1 Introduction we will mention the scalability challenges associated with federated learning in large-scale networks, and highlight that the framework employs a P2P aggregation strategy to mitigate potential bottlenecks.
In Section 2 Materials and Methods, Subsection 2.5 (the newly added section as explained in response to comment 1), we provide a more detailed explanation of the decentralized P2P aggregation process, emphasizing how it eliminates the reliance on a central server and distributes the computational and communication load. And we discuss how the P2P approach contributes to improved scalability by avoiding single points of failure and enabling parallel processing of model updates.
- There is no discussion on Integration challenges, such as (a) Interoperability (interaction of the proposed framework with different EHR systems); (b) data standards (just mentioning about the integration issue with existing healthcare systems is not enough, the authors need to discuss the need for standard data models and ontologies to ensure seamless integration)
Response: We agree with the reviewer that it is necessary to highlight the critical importance of integration for the success of our collaborative framework. While the primary focus of this paper is on distributed learning and security, including privacy, we recognize that seamless integration with existing healthcare systems is essential for real-world adoption.
In our previous work [1-4], we have actively explored the use of ontologies and knowledge graphs to address integration and interoperability challenges in healthcare data management. We believe that leveraging these techniques within our current framework can significantly enhance its ability to interact with diverse EHR systems and ensure consistent data representation.
To address this valuable feedback, we have revised the manuscript in the following ways:
In Section 2 3. Materials and Methods, at the end of the first part, we have introduced a brief overview of our previous work on integration and interoperability using ontologies and knowledge graph, highlighting their potential relevance to the current framework.
References:
- Sarani Rad, Fatemeh, Rasha Hendawi, Xinyi Yang, and Juan Li. 2024. "Personalized Diabetes Management with Digital Twins: A Patient-Centric Knowledge Graph Approach" Journal of Personalized Medicine 14, no. 4: 359. https://doi.org/10.3390/jpm14040359
- Rasha Hendawi and Juan Li, Comprehensive Personal Health Knowledge Graph for Effective Management and Utilization of Personal Health Data, IEEE AIMHC 2024 IEEE First International Conference on Artificial Intelligence for Medicine, Health and Care
- Vikram Pandey, Juan Li and Shadi Alian, “Evaluation and Evolution of NAOnto – An Ontology for Personalized Diabetes Management for Native Americans”, 2021 7th IEEE International Conference on Computer and Communications (ICCC) Chengdu, China, December 10-13, 2021
- Rasha Hendawi, Shadi Alian and Juan Li, “ Breaking Down Barriers: Empowering Diabetes Patients with User-Friendly Medical Explanations” The 15th IEEE International Conference on Information and Communication Systems (ICICS2024)
- Implementation mechanisms are mentioned but not discussed properly. Therefore, a detailed description of the security measures (such as encryption techniques, threat mitigation strategies, access control, etc.) and smart contracts (how the design and implementation of smart contracts for model updates, data access control, incentive mechanisms, etc. will be carried out) needs to be provided.
Response: We acknowledge that a clear and comprehensive explanation of the security measures and smart contract implementation is crucial for understanding the practical aspects of our proposed solution and ensuring its robustness in real-world scenarios. We need to point out that for smart contracts, we have explained the detailed implementation with examples in the use cases section.
To address this concern, we propose the following revisions to the manuscript:
In Section 2 Materials and Methods’ subsection 2.2 Privacy Preservation, we expand the discussion on encryption techniques, providing more specific examples and explanations of how they are applied to protect sensitive model updates during the aggregation process.
Elaborate on the advantages and limitations of different encryption techniques, such as homomorphic encryption and secure multi-party computation, in the context of federated learning and blockchain. We provide a more detailed explanation of the data anonymization techniques being explored, highlighting their potential to further enhance privacy while preserving data utility.
In Section 2.3 Secure Collaboration, we provide a more in-depth description of the smart contract architecture and its key components. We explain the specific design and implementation details of smart contracts for:
- Model updates: How model updates are securely submitted, validated, and aggregated on the blockchain.
- Data access control: How access permissions are defined, enforced, and audited using smart contracts.
- Incentive mechanisms: How rewards and incentives are distributed to participants using transparent and auditable smart contract logic.
More smart contract use case examples are presented in Section 3.3.2. Blockchain Smart Contract Evaluation with Use cases
In Section 4 Discussion, we expand the discussion on security measures to include specific threat mitigation strategies, such as:
- Input validation and sanitization to prevent malicious data injection.
- Anomaly detection mechanisms to identify and address potential security breaches or suspicious activities.
- Resilience against blockchain-specific attacks like 51% attacks or double-spending.
- Some sections miss clear explanations and need a more structured presentation of the methodologies and findings, such as (a) the introduction section: fails to establish the novelity of their proposed framework compared to the existing state of the art; (b) the methodology section: needs to provide a step-by-step explanation of the implementation of the framework. The most important missing components are: specific algorithms, data preprocessing steps, model training processes, and the exact role of federated learning and blockchain in the proposed framework, to name a few.
Response: (a) We acknowledge that the initial introduction did not adequately highlight the unique aspects of our approach compared to existing work. To rectify this, we have revised the introduction (Section 1) to explicitly state how our framework distinguishes itself by synergistically integrating blockchain and federated learning to address the specific challenges of diabetes prediction in a decentralized, secure, and privacy-preserving manner. We also emphasize the use of a peer-to-peer (P2P) aggregation strategy to enhance scalability and efficiency in federated learning.
(b) We appreciate the reviewer's feedback regarding the need for a more detailed, step-by-step explanation of the implementation in the methodology section. We understand the importance of clarity and reproducibility, especially when presenting a new framework. However, our primary intention in this paper is to introduce a general framework that can serve as a reference for decentralized and secure healthcare learning and analysis, rather than focusing on a specific implementation. Therefore, the methodology section primarily explains the overarching concepts and design principles, providing a high-level overview of the framework's architecture and key components.
We did provide the specific implementation details, including the choice of algorithms, data preprocessing steps, model training processes, in the Results section.
By keeping the methodology section focused on the general framework, we enhance its adaptability and applicability to a broader range of healthcare use cases and data types. The Results section serves as a showcase of a concrete prototype implementation and evaluation, offering readers a tangible example of how the framework can be applied in practice. This approach allows us to demonstrate the feasibility and effectiveness of our approach while maintaining the general nature of the framework itself.
In response to the reviewer's feedback, we have reorganized and rewritten the methodology section to improve clarity and explicitly point readers to the Results section for specific implementation details. We have pointed out the exact role of blockchain and federated learning, We have also added subheadings and numbered lists to enhance readability and organization.
We believe that this approach strikes a balance between presenting a generalizable framework and providing sufficient implementation details to demonstrate its practical application.
Reviewer 2 Report
Comments and Suggestions for Authors
The paper proposes a blockchain-based Collaborative Framework for Personalized Diabetes Prediction to provide high-quality services such as security and data privacy. To evaluate the trustworthiness of the proposed system, the authors state that they simulated the blockchain networks promising solution for the ethical and effective use of healthcare data in diabetes prediction.
1. The paper is rather unbalanced. Although the evaluation section presents promising results, significantly more information is required regarding the methodology and experimental setup. There is virtually no information within the paper on how the experiment and simulation were setup.
2. Also the related work section reads very much like a continuation of the introduction with relatively few references. The number of references is sufficient, even if some discussions about the current state of the art could be added. See for example: i) A Decentralized Resource Allocation in Edge Computing for Secure IoT Environments and ii) Blockchain-based trust mechanism for digital twin empowered Industrial Internet of Things
3. Then comparing these works, and highlighting the differences with the contributions in this paper. Therefore section 3.3 needs to be substantially expanded.
4. Figure 8: Healthcare provider is granted access to patient data - figure 12 Malicious model updates are avoided and must be presented with better clarity and explanation.
5. Overall I commend the authors and researchers for their work, but substantial improvements need to be made in writing up this work into a research paper that communicates all elements.
Author Response
- The paper is rather unbalanced. Although the evaluation section presents promising results, significantly more information is required regarding the methodology and experimental setup. There is virtually no information within the paper on how the experiment and simulation were setup.
Response: We appreciate the reviewer's valuable feedback regarding the need for more comprehensive information on the methodology and experimental setup. We recognize that a clear and detailed explanation of these aspects is crucial for understanding the technical intricacies of our framework and ensuring the reproducibility of our results.
In response to this feedback, we have made significant revisions to the "Results" section (Section 3) of the manuscript. We have expanded the section to include a dedicated subsection on "Experimental Setup" (Section 3.1), which provides a step-by-step explanation of the following:
- Blockchain Network Configuration: We detail the specific blockchain platform used (Ethereum), the number of participating nodes, and the development environment employed for smart contract implementation.
- Dataset Preparation: We describe the source of the dataset, the data partitioning strategy to simulate a distributed environment, and the specific preprocessing steps applied to ensure data quality and consistency.
- Model Selection and Training: We clearly state the chosen machine learning algorithm (XGBoost) and provide a brief justification for its selection. We also elaborate on the hyperparameter optimization process using Optuna.
- Federated Learning Workflow: We provide a detailed walkthrough of the federated learning process, explaining each step from model initialization to iterative training, secure aggregation, and model distribution.
Furthermore, we have enriched the "Evaluation Results" subsection (Section 3.3) to include:
- Specific Performance Metrics: We explicitly state the key metrics used to evaluate the model's performance, including accuracy, precision, recall, and F1-score.
- Convergence Analysis: We discuss the convergence behavior of the federated learning model, highlighting the improvements in performance metrics over training rounds and the point of convergence.
- Comparison with Centralized Learning: We present a comparison of the communication overhead between our federated learning approach and centralized learning, demonstrating the significant reduction in communication costs.
Finally, the "Blockchain Smart Contract Evaluation with Use Cases" subsection (Section 3.3.2) provides concrete examples to showcase the practical implementation and effectiveness of the smart contracts in ensuring data privacy and security.
We believe that these revisions have significantly enhanced the clarity and comprehensiveness of the "Results" section, providing a more detailed and transparent explanation of our experimental setup and methodology. We are confident that the revised manuscript now adequately addresses the reviewer's concerns and enables a better understanding of the technical aspects of our work.
- Also the related work section reads very much like a continuation of the introduction with relatively few references. The number of references is sufficient, even if some discussions about the current state of the art could be added. See for example: i) A Decentralized Resource Allocation in Edge Computing for Secure IoT Environments and ii) Blockchain-based trust mechanism for digital twin empowered Industrial Internet of Things
Response: We understand the suggestion to expand the discussion and include references to related work in other domains, such as the examples provided (i.e., decentralized resource allocation in edge computing and blockchain-based trust mechanisms for digital twins in IoT). However, we respectfully point out that the organization of this paper strictly adheres to the Biomedicines journal's requirements, which do not include a dedicated "Related Work" section. As such, we have integrated the discussion of relevant literature into the Introduction, focusing primarily on prior research in diabetes management and healthcare applications of blockchain and federated learning. This approach allows us to contextualize our work within the specific domain of diabetes prediction while adhering to the journal's formatting guidelines.
Nevertheless, we acknowledge the value of drawing connections to related work in other fields. To address the reviewer's suggestion, we have expanded the Introduction to include a brief discussion on the broader applicability of our framework beyond diabetes prediction. We highlight the potential for our approach to be adapted to other healthcare domains where privacy, security, and data diversity are critical concerns. We also briefly mention the relevance of our work to ongoing research in decentralized resource allocation and trust mechanisms in edge computing and IoT environments, demonstrating the potential for cross-domain knowledge transfer and collaboration.
We believe that these revisions strike a balance between adhering to the journal's formatting requirements and providing a broader context for our research. We are confident that the expanded discussion in the Introduction adequately situates our work within the existing literature while highlighting its potential impact beyond the specific domain of diabetes prediction.
- Then comparing these works, and highlighting the differences with the contributions in this paper. Therefore section 3.3 needs to be substantially expanded.
Response: We appreciate the reviewer's suggestion to include a more in-depth comparison of our work with existing state-of-the-art approaches in the Results section (Section 3.3). While we recognize the value of comparative analysis, we would like to respectfully point out that to the best of our knowledge, there is currently no highly related work that directly combines blockchain and federated learning for personalized diabetes prediction using a similar methodology and dataset.
The examples provided by the reviewer (decentralized resource allocation in edge computing and blockchain-based trust mechanisms for digital twins in IoT) are indeed relevant in terms of the underlying technologies and concepts. However, their application domains, datasets, and specific use cases differ significantly from our focus on diabetes prediction within the healthcare context. Therefore, a direct quantitative or qualitative comparison may not be entirely meaningful or informative for the readers of this paper.
Nevertheless, we have taken the reviewer's feedback into consideration to include a more comprehensive discussion of the unique contributions and advantages of our framework. We have highlighted the key distinctions between our approach and existing works in terms of the synergistic integration of blockchain and federated learning, the emphasis on patient privacy and regulatory compliance, and the use of a decentralized P2P aggregation strategy for enhanced scalability. We believe that this expanded discussion provides a clearer understanding of the novelty and potential impact of our research within the specific context of diabetes prediction.
We also acknowledge that as the field of blockchain and federated learning in healthcare evolves, new approaches and methodologies may emerge that offer more direct points of comparison. We are committed to staying abreast of these developments and incorporating relevant comparative analyses in future iterations of our work.
- Figure 8: Healthcare provider is granted access to patient data - figure 12 Malicious model updates are avoided and must be presented with better clarity and explanation.
Response:We acknowledge that visual representations play a crucial role in conveying complex concepts effectively. We have carefully addressed this concern by creating new screenshot figures that incorporate additional context and provide more comprehensive explanations.
Round 2
Reviewer 1 Report
Comments and Suggestions for Authors
I am satisfied with your answers and the changes that you have made to the article. Thanks
Reviewer 2 Report
Comments and Suggestions for Authors
Accept